# Sarcopenia Screening Allows Identifying High-Risk Patients for Allogenic Stem Cell Transplantation

**DOI:** 10.3390/cancers13081771

**Published:** 2021-04-08

**Authors:** Johannes Kirsten, Verena Wais, Sebastian V.W. Schulz, Elisa Sala, Gunnar Treff, Donald Bunjes, Jürgen M. Steinacker

**Affiliations:** 1Center for Internal Medicine, Division of Sports and Rehabilitation Medicine, Ulm University Medical Center, 89073 Ulm, Germany; Sebastian.Schulz@uniklinik-ulm.de (S.V.W.S.); Gunnar.Treff@uniklinik-ulm.de (G.T.); Juergen.Steinacker@uniklinik-ulm.de (J.M.S.); 2Unit for Allogenic Blood Stem Cell and Bone Marrow Transplants, Clinic for Internal Medicine III, Center for Internal Medicine, Ulm University Medical Center, 89073 Ulm, Germany; verena.wais@uniklinik-ulm.de (V.W.); Elisa.Sala@uniklinik-ulm.de (E.S.); Donald.Bunjes@uniklinik-ulm.de (D.B.)

**Keywords:** sarcopenia, hematology, risk stratification, aerobic capacity, stem cell transplantation

## Abstract

**Simple Summary:**

Allogenic stem cell transplantation is a treatment option for various hematological diseases. Due to the intensity of the therapy regimes used, there is a substantial therapy associated mortality and morbidity. Therefore, it is crucial to identify patients with increased risk for treatment associated complications. Sarcopenia, defined as the loss of muscle mass and strength is a risk factor in various diseases. Aim of our study was to implement and evaluate the predictive power of a sarcopenia assessment, based on muscle mass, muscle strength and aerobic capacity (by measuring peak oxygen uptake), on all-cause and non-relapse mortality. A total of 178 patients were screened, with 28% suffering from sarcopenia before transplantation. Our results show a three-fold increase in all-cause and non-relapse mortality in this subpopulation compared to non-sarcopenic patients within a 12-month follow up. The importance of physical performance status demonstrated, raises the question, if exercise interventions might even allow to decrease mortality and morbidity.

**Abstract:**

Allogenic stem cell transplantation (aSCT) is the only potentially curative treatment for high-risk hematological diseases. Despite advancements in supportive measures, aSCT outcome is still affected by considerable transplant-related mortality. We implemented a new sarcopenia assessment prior to aSCT to evaluate its predictive capability for all-cause and non-relapse mortality. Therefore all patients initially scheduled for aSCT within a 25-month period were screened during pre-transplantation-routine for muscle mass, grip strength, and aerobic capacity (AC) by measuring peak oxygen uptake (VO2peak). Patients were assigned to one of five groups adapted according current sarcopenia guidelines. Primary endpoints were all-cause and non-relapse mortality within a follow up time of up to 12 months. A total of 178 patients were included and rated as normal (*n* = 48), impaired aerobic capacity (*n* = 56), pre-sarcopenic (*n* = 26), sarcopenic (*n* = 27), and severe sarcopenic (*n* = 22) without significant age-differences between groups. Patients presenting with sarcopenia showed a significant three-fold increase in all-cause and non-relapse mortality compared to patients with normal screening results. AC showed to be the strongest single predictor with a more than two-fold increase of mortality for low AC. We conclude that risk stratification based on combination of muscle mass, grip strength, and AC allowed identifying a subgroup with increased risk for complications in patients undergoing aSCT.

## 1. Introduction

For high-risk hematological diseases, allogenic stem cell transplantation (aSCT) is the only potentially curative treatment option, in some cases representing the last line therapy in a long treatment history (e.g., for Myeloma patients and some lymphoma patients, [1]. Despite optimization of supportive therapy regimes, there are still substantial treatment related morbidity and mortality associated with this procedure, with one-year overall mortality still reaching 25% [2]. Thus, risk stratification for transplantation-associated complications is of high clinical importance when selecting patients, also in order to potentially adjust conditioning regimens and subsequent treatments to the individual [3].

Two main score systems are actually widely used in order to estimate the non-relapse mortality for patients undergoing aSCT, thus helping the decisional process. The hematopoietic cell transplantation specific comorbidity index (HCT-CI) is widely established as a tool to assess patient comorbidities for risk stratification. The EBMT-Score considers other variables, like disease stage, previous treatments and type of donor as well as patients’ age. It has been shown that HCT-CI values >2 a.U. as well as EBMT values >1 a.U. are associated with increased mortality [4,5,6,7,8]. However, in the age of precision medicine, advanced and customized risk stratification allowing building individual treatment strategies, is still an unmet clinical need. This yields especially in the field of aSCT, where—with the emergence of conditioning regimens of reduced intensity and advances in supportive therapy—a substantial proportion of older patients (>60 years) are being evaluated for intensive treatment options. In this context, evaluation of physical performance could play a crucial role in individualizing treatment strategies.

Sarcopenia, defined by the European working group on sarcopenia (EWGSOP) as “a progressive and generalized skeletal muscle disorder with increased likelihood of adverse outcomes” [9,10], has been shown to be associated with a decreased overall survival in cancer patients, e.g., in acute myeloid leukemia [11]. In the context of sarcopenia, muscle mass, strength and physical performance are attenuated in an interdependent manner. Consequently sarcopenia assessment necessitates measurements of muscle strength, muscle mass and physical performance, with strength, rather than muscle mass, recently proving to be the most important component [10]. To date studies on prevalence of sarcopenia in cancer patients and its predictive value for overall survival often focused on muscle mass measurement without assessment of strength or physical performance [12]. Furthermore, cardio pulmonary exercise testing (CPET) has not been implemented into sarcopenia screenings yet, even though CPET allows assessing aerobic capacity (AC) via peak oxygen consumption (VO_2peak_). VO_2peak_ represents an overall measure of the cardio pulmonary system, mirroring the complex product of ventilation, perfusion and diffusion in the lung, oxygen transport capacity of heart and blood, and the peripheral muscles’ oxidative capacity [13], thus serving as a surrogate for overall cardio-pulmonary performance. Feasibility of CPET in patients before aSCT has been shown by Kelsey et al. [14]. and VO_2peak_ has proven to be predictive for overall survival in a variety of diseases [15,16,17]. Thus far, to the best of our knowledge, no studies included VO_2peak_ into sarcopenia screenings in patients awaiting aSCT.

To this end, we aimed to evaluate the predictive power of a sarcopenia screening that combines measurements of muscle strength, muscle mass, and AC on all-cause and non-relapse mortality in patients scheduled for aSCT, in order to potentially identify patients with high-risk for all-cause and non-relapse mortality.

## 2. Materials and Methods

All patients scheduled for aSCT between April 2018 and Mai 2020 in the Unit of Bone Marrow Transplantation of the Hematological Department of University Hospital of Ulm, Germany, were enlisted to this prospective observational study. The approval for this study was obtained from the ethical board of Ulm University (339/18). Written informed consent was obtained from all subjects before the enrollment in the study. This trial was registered in the German register of clinical trials (DRKS00021425) and was carried out according to the declaration of Helsinki. Once indication and eligibility for the aSCT were established, all patients underwent an additional assessment of sarcopenia status as follows.

### 2.1. Muscle Strength Testing

In line with similar research, muscle strength was assessed by measuring grip strength [18,19,20] that was measured in triplicate on both hands using a hand dynamometer (SH1003, Saehan Corp., Donghae, Korea). The highest value was used for the classification according to age and sex related normal values from a large European population [21]. The 25th percentile equaling one standard deviation of the normal values was used as a cutoff to discriminate normal from pathological low grip strength. To compare results of different age and sex, individual results were standardized calculating a Z-Score, with the 25th percentile of the individual normal value, corresponding to a Z-score of −0.68.

### 2.2. Body Composition

Body composition was analyzed using bioelectrical impedance technology (InBody 770, Biospace Korea, Seoul, Korea). Total muscle mass was adjusted to body height and values below 10.76 kg/m^2^ in men and below 6.76 kg/m^2^ in women were rated as pathologically low [22,23].

### 2.3. Short Physical Performance Battery

To allow for a comparison of physical performance status to a test already established in sarcopenia screenings, the Short Physical Performance Battery was applied. Results ≤ 8 points were defined as pathologically low [24,25].

### 2.4. Cardiopulmonary Exercise Testing

All patients underwent CPET to assess their AC by measuring VO_2peak_, combined with electrocardiogram (AMEDTEC Cardiopart 12B, AMEDTEC Medizintechnik Aue GmbH, Aue, Germany) on a cycle ergometer (Lode Excalibur Sport, Lode B.V., Groningen, The Netherlands). A ramp wise incremental test protocol (25 W + 15 W/min) till voluntary exhaustion was used to assess VO_2peak_ during CPET utilizing a breath-by-breath metabolic analyzer (Ergostik, Geratherm, Geratal, Germany). VO_2peak_ values below 80% of the individual normative values adjusted to sex, age, body mass, and height using the Hansen/Wasserman equations [26] were rated as “low”. VO_2peak_ was defined as the highest 30-s rolling average with a respiratory exchange ratio (RER) ≥ 1.10. In *n* = 48 patients who failed to reach cardiopulmonary exhaustion, ventilatory threshold 1 was assessed and results below 80% of the normative values were rated as “low” AC [27].

### 2.5. Assignment to the Corresponding Sarcopenia Category

The following categories have been adapted in line with the original EWGSOP consensus (Cruz-Jentoft et al., 2010).

Normal: All patients with normal values in each conducted test were assigned to this category.

Impaired aerobic capacity: All patients with low AC only, while having normal muscle strength and muscle mass, were assigned to this category.

Pre-sarcopenia: Patients either presenting with low grip strength or low muscle mass while having normal aerobic capacity were considered to be pre-sarcopenic.

Moderate sarcopenia: Patients with either low grip strength or low muscle mass, and low AC were assigned to this category.

Severe sarcopenia: Patients with low grip strength, low muscle mass and low AC were considered to be severe sarcopenic.

### 2.6. Scores

HCT-CI/age and DRST-EBMT scores were obtained during the routine data assessment in the Unit for Allogenic Blood Stem Cell and Bone Marrow Transplants at University Hospital Ulm, Germany.

### 2.7. Statistics

Primary outcome was all-cause mortality, defined as the number of days surviving after initial screening by each patient. Patients were observed until death or 12 month when they were censored.

Continuous variables are given as mean ± standard deviation. Categorical variables are presented as absolute and percentage frequencies, respectively.

To evaluate differences between groups 1–4, unpaired t-tests were applied. Differences in categorical variables were evaluated using Χ^2^-test or one-way ANOVA where appropriate. For survival analysis, Kaplan–Meier method and the log-rank test were used. To analyze potential bias in all-cause and non-relapse mortality caused by the applied conditioning regime, differences for the categories of the model and the predictors were calculated using cox regression with the regime as covariate.

A two-sided *p*-value of less than 0.05 was considered to be statistically significant. Due to the nature of this study, all results from statistical tests have to be regarded as explorative. Statistical analyses were conducted with GraphPad prism 8 (GraphPad Software, San Diego, CA, USA) and SPSS (IBM Corp. Released 2017. IBM SPSS Statistics for Windows, Version 25.0. IBM Corp., Armonk, NY, USA).

## 3. Results

### 3.1. Patients and Screening

A total of 199 patients was consecutively screened between 2018 and 2020. Sarcopenia screening was conducted on average 1.06 month before transplantation. Twenty-one patients were excluded from analysis due to screening after transplant (*n* = 2), change of therapy regime (*n* = 15), or withdrawal of consent (*n* = 4). The remaining patients (*n* = 178) were assigned to five different sarcopenia categories (Figure 1).

A subset of 48 patients (27%) was assigned to group 1 (normal), 55 patients (31%) to group 2 (impaired aerobic capacity), 26 patients (15%) to group 3 (pre-sarcopenic), and 49 patients (28%) were pooled in group 4 (sarcopenic), of whom 27 patients (15%) were categorized with moderate and 22 patients (12%) with severe sarcopenia. No patient missed more than one part of the examination.

Patients’ baseline characteristics and diagnosis are shown in Table 1. Follow-up time was limited to 12 months, with the average follow up time of survivors being 11.33 months (until 31 December 2020) without significant differences between groups. Neither age was significantly different between groups nor the distribution of diagnoses necessitating aSCT.

Eight patients of groups 1–4 (*n* = 1; *n* = 3; *n* = 1; *n* = 3) died prior to transplantation. 170 patients were transplanted as scheduled.

From patients transplanted 116 developed an acute Graf-versus-Host-Disease (aGvHD) and 44 a chronic Graft-versus-Host-Disease (cGvHD) during the follow-up time. Again incidence of aGvHD or cGvHD was not significantly different between the groups.

There was no significant difference in length of hospital stay for the actual transplantation between the groups.

### 3.2. All-Cause Mortality

At the time of data cutoff 57 patients (32%) had passed, with overall 12-month all-cause mortality amounting to 18.75% in group 1, 32.73% in group 2, 30.77% in group 3, and 44.9% in group 4 (for main reasons of mortality see Appendix A).

A statistically significant trend for increasing all-cause mortality was found from group 1 to group 4 (log-rank test for trend *p* = 0.0062; Figure 2A). When comparing group 4 (i.e., sarcopenic and severe sarcopenic) with group 1 (normal), HR was 3.12 (95% CI [1.54, 6.35] *p* = 0.002).

Differences in all-cause mortality were calculated for each of the screening’s components separately (i.e., muscle mass, grip strength, AC) resulting in a significant HR for low aerobic capacity (HR 2.16 (95% CI [1.28, 3.67] *p* = 0.01), Figure 2D). Differences in all-cause mortality for muscle mass and grip strength were not significant, although indicating a trend for increasing all-cause mortality for low muscle mass and low grip strength (HRs 1.65 (95% CI [0.90, 3.01]) *p* = 0.09, Figure 2B and 1.56 (95% CI [0.87, 2.80]) *p* = 0.09, Figure 2C).

When combining all screening components using cox-regression the HR for AC remained significant (HR 1.94 (95% CI [1.04, 3.63]) *p* = 0.04).

During the pre-transplantation workup, HCT-CI/age was determined in 171 patients. No significant difference in all-cause mortality was found for HCT-CI/age >2 (HR 1.29 (95% CI [0.70, 2.40]) *p* = 0.42). Of note, only 42 patients presented with an HCT-CI/age >2 in our patient population.

In the short physical performance battery, all but one patient had a result higher than eight points.

### 3.3. Non-Relapse Mortality

One hundred and seventy patients were transplanted as scheduled. Total 12-month non-relapse mortality was *n* = 38. A significant trend for lower non-relapse mortality was found from group 1 to group 4 (log-rank test for trend *p* = 0.03; Figure 3A). When comparing group 4 (i.e., sarcopenic and severe sarcopenic) with group 1 (normal) HR was statistically significant, being 2.97 (95% CI [1.23, 7.19] *p* = 0.02).

Differences in non-relapse mortality were also calculated for each of the screening’s components separately (i.e., muscle mass, grip strength, AC) resulting in a significant HR for low AC (HR 2.21 (95% CI [1.11, 4.43] *p* = 0.04), Figure 3D). Differences in non-relapse mortality for muscle mass and grip strength were not significant although indicating a trend for increasing non-relapse mortality for low muscle mass and low grip strength (muscle mass HR 1.52 (95% CI [0.69, 3.35] *p* = 0.36) Figure 3B; grip strength HR 1.73 (95% CI [0.80, 3.71] *p* = 0.15) Figure 3C).

### 3.4. Conditioning Regime, Overall Survival, and Non-Relapse Mortality

In the 170 patients transplanted either a reduced intensity conditioning regime (*n* = 113) or a myeloablative conditioning regime (*n* = 57) was applied.

When adjusted for the conditioning regime, HRs were significantly higher for Group 4 (i.e., sarcopenic and severe sarcopenic) vs. group 1 (normal) with HR of 3.13 (95% CI [1.35, 7.3] *p* = 0.008 for all-cause mortality and 2.94 (95% CI [1.10, 7.83] *p* = 0.03 for non-relapse mortality, respectively. HRs for low AC remained significant for all-cause mortality (HR 1.99 95% CI [1.05, 3.75] *p* = 0.03) and non-relapse mortality (HR 2.19 95% CI [0.99, 4.86] *p* = 0.05) confirming the previous results.

The HRs for low muscle mass and low grip strength were not statistically significant for predicting all-cause mortality (low muscle mass: HR 1.52 (95% CI [0.85, 2.75]) *p* = 0.16, low grip strength: HR 1.31 (95% CI [0.72, 2.36]) *p* = 0.37) and non-relapse mortality (muscle mass: HR 1.38 (95% CI [0.67, 2.85]) *p* = 0.38, grip strength: HR 1.53 (95% CI [0.76, 3.08]) *p* = 0.23) when adjusted for the conditioning regime.

## 4. Discussion

This is the first prospective study indicating that a sarcopenia assessment combining measurement of strength, muscle mass and AC allows identifying a high-risk population for the endpoints all-cause mortality and non-relapse mortality, within 178 patients awaiting aSCT. The assessment showed high predictive capability, as indicated by a three-fold higher all-cause and non-relapse mortality in sarcopenic compared to normal patients (groups 4 vs. 1), notably even when adjusted for the conditioning regime.

Our proposed categorization model derived from the diagnostic criteria for sarcopenia in the geriatric setting identified 27.5% of the patients as being sarcopenic or severe sarcopenic and, therefore, at high-risk for the endpoints investigated.

There was no significant accumulation of specific diagnoses within any of the groups, indicating that the differences in previous therapies did not disproportionately drive the allocation to a certain group. Furthermore, there were no significant age differences between groups, indicating that sarcopenia is not limited to a particular age in hematologic patients.

In contrast to the results obtained with our innovative screening, only few comorbidities were found (HCT-CI/age 0–2). Recalling that patients with high HCT-CI/age values have already been excluded from aSCT beforehand, HCT-CI/age values do obviously not allow for a further risk-stratification in the remainder, i.e., those who are scheduled for aSCT. Potentially useful performance markers of the HCT-CI/age like echocardiography or lung function testing are conducted at rest.

It is worth to mention that a large number of patients in our study developed an acute Graft-versus-Host-Disease (aGvHD). aGvHDis often treated with corticosteroids, potentially inducing myopathy that may worsen the situation of already sarcopenic patients. However, there was neither an accumulation of aGvHD within a certain sarcopenia group nor among patients that passed, hence, there is no indication that corticosteroid associated myopathy confounded our data.

### Components of the Screening

In our study, muscle mass was rated to be low in 26.4% of the patients and failed to predict all-cause or non-relapse mortality significantly, although two studies utilizing retrospective analysis of CT scans have shown a significant increase of all-cause mortality with low muscle mass [11,28]. However, recent data as well as our results indicate that muscle function is more important for the outcome, a conclusion which has also been included into the current EWGSOP2-statement 2019 [10].

However, grip strength alone also failed to predict all-cause and non-relapse mortality in our patient population significantly. Of note, the 25th percentile of grip strength from the normative dataset derived from a British population by Dodds et al. [21] was used as cutoff point, because this dataset is based on 49,964 participants (26,687 female) aging 4–90 years, which is an important difference to the EWGSOP guidelines that solely cover geriatric patients. In contrast, grip strength was a predictor of all-cause mortality in the aSCT setting in a recent study by Salas et al. [29] who reported a significant HR with a 1.6-fold higher mortality similar to our results for low grip strength without adjusting the cut-offs to patients’ age.

AC was found to be the strongest single predictor of all-cause and non-relapse mortality in our study, also when adjusted for the used conditioning regime. A low AC was associated with a more than two-fold higher (non-relapse) mortality. This is not surprising, since AC is an integral measure including ventilation and perfusion, oxygen transport capacity, and peripheral muscle oxidative capacity.

The validity of our findings is underpinned by the fact that 92% of all patients who were scheduled for aSCT in our hospital during the study period were included in the study. We have therefore avoided selection bias as far as possible, which is a common limitation in similar studies [14,30].

Aside from the superior diagnostic and predictive capability of the screening, we also demonstrated the feasibility of the screening routine. There were no incidents caused by the screening. Especially, exercise testing until voluntary exhaustion proved to be safe within this patient population.

After integration in the regular pre-transplant workup, the sarcopenia assessment was performed in approx. 2 h. We are aware, that additional resources are needed to implement such procedures in transplantation centers, but the effort appears to be valuable in light of the expenses of peri-transplant complications.

## 5. Conclusions

We conclude that the combined evaluation of muscle mass, strength, and AC is feasible and allows for the identification of patients who need special attention and tailored training during and after the hospitalization period and may benefit from conditioning regimes of reduced intensity. The influence of physical performance status on all-cause and non-relapse mortality, also reported by several other studies, raises the question, if timely exercise interventions, possibly complemented by elevated protein intake, might even allow decreasing mortality and morbidity in patients undergoing aSCT.

In addition, the integration of such screening routine into the pre-transplantation workup might enable clinicians to prescribe individual and appropriate training programs for each patient, allowing to reduce the selection bias of several exercise studies in cancer patients, where patients of poor performance status and high burden of disease do not sign up for training at all. Therefore, prospective multi-center studies with larger sample sizes, focusing especially on patients of low performance status, are urgently needed to optimize exercise interventions promising to reduce mortality without attenuating the therapy’s potential.

## Figures and Tables

**Figure 1 cancers-13-01771-f001:**
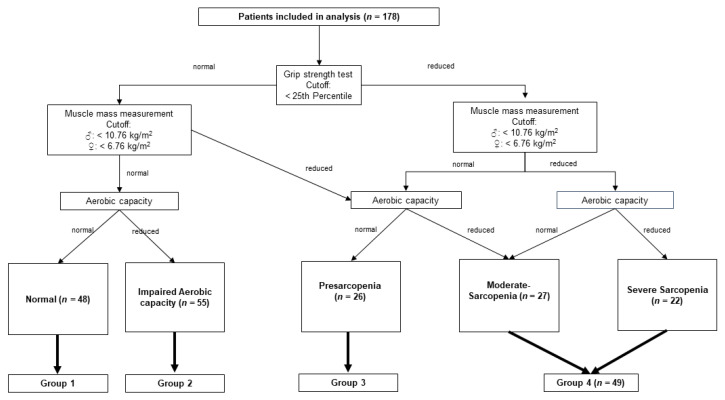
Schematic overview of a sarcopenia screening in 178 patients awaiting allogenic stem cell transplantation (aSCT) and their assignment to one of five categories representing their individual sarcopenia status, according to the results of grip strength testing, muscle mass measurement and cardiopulmonary exercise testing for assessing aerobic capacity. Note: Patients with sarcopenia and severe sarcopenia were grouped for analysis (group 4).

**Figure 2 cancers-13-01771-f002:**
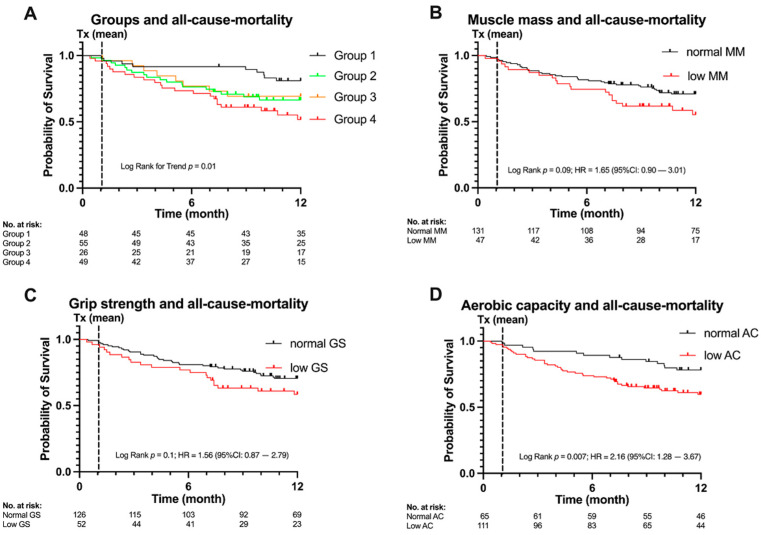
All-cause mortality, categories and predictors (**A**) Group 1 (normal), group 2 (impaired aerobic capacity (AC)), group 3 (presarcopenic), group 4 (sarcopenic/severe sarcopenic). Significant trend for increasing all-cause mortality (*p* = 0.006) with increasing sarcopenia status. (**B**) Kaplan–Meier-Plot for muscle mass (MM), hazard ratio (HR) 1.65 (95% CI [0.90, 3.01]) *p* = 0.09. (**C**) Kaplan–Meier-Plot for grip strength (GS), HR 1.56 (95% CI [0.87, 2.80]) *p* = 0.09. (**D**) Kaplan–Meier plot for low aerobic capacity (AC), HR 2.16 (95% CI [1.28, 3.67] *p* = 0.01. Time 0 is the day of the sarcopenia assessment, mean day of transplantation as indicated by Tx is 1.06 months.

**Figure 3 cancers-13-01771-f003:**
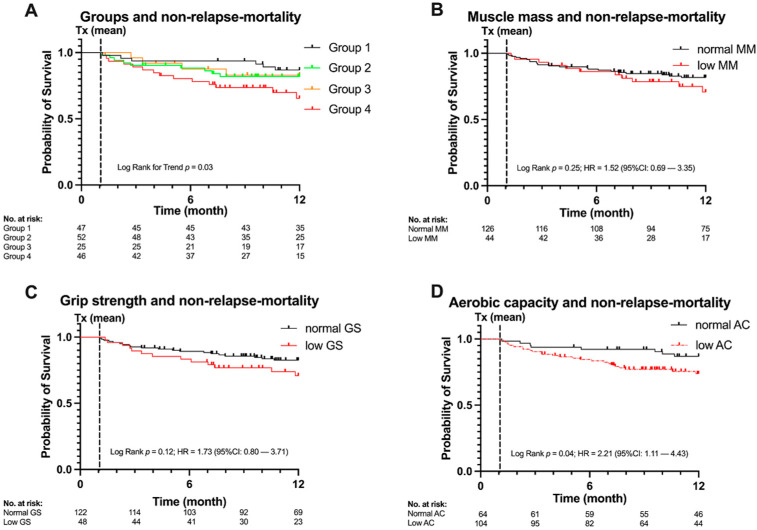
Non-relapse mortality, categories and predictors. (**A**) Group 1 (normal), group 2 (impaired aerobic capacity (AC)), group 3 (pre-sarcopenic), group 4 (sarcopenic/severe sarcopenic). A significant trend for increasing non-relapse mortality (*p* = 0.03) with increasing sarcopenia status. (**B**) Kaplan-Meier-Plot for muscle mass (MM) and non-relapse mortality, hazard ratio (HR) 1.52 (95% CI [0.69, 3.35] *p* = 0.36. (**C**) Kaplan–Meier plot for grip strength (GS) and non-relapse mortality, HR 1.73 (95% CI [0.80, 3.71] *p* = 0.15). (**D**) Kaplan-Meier-Plot for low aerobic capacity (AC) and non-relapse mortality, HR 2.21 (95% CI [1.11, 4.43] *p* = 0.04). Time 0 is the day of the sarcopenia assessment, mean day of transplantation as indicated by Tx is 1.06 months.

**Table 1 cancers-13-01771-t001:** Patients’ characteristics and results in the different analysis groups.

		Group 1	Group 2	Group 3	Group 4	Total
		(Normal)	(Impaired)	(Pre-Sarcopenic)	(Sarcopenic)	
			Aerobic Capacity			
Age (Years)		59.0 (27–71.9)	57.3 (18.8–83.7)	59.4 (19.1–70.6)	56.9 (18.7–70.8)	58.3 (18.7–83.7)
Sex (*n*)	male	20 (18.3%)	30 (27.5%)	17 (16.6%)	42 (38.5%)	109
	female	28 (40.6%)	25 (36.2%)	9 (13.0%)	7 (10.1%)	69
	all	48 (27.0%)	55 (30.9%)	26 (14.6%)	49 (27.5%)	178
Diagnosis (*n*)	MDS	13 (34.2%)	14 (36.8%)	4 (10.5%)	7 (18.4%)	38
	AML	16 (23.2%)	17 (24.6%)	13 (18.8%)	23 (33.2%)	69
	ALL	2 (15.4%)	3 (23.1%)	3 (23.1%)	5 (38.5%)	13
	CML	4 (50.0%)	2 (25%)	2 (25%)	0 (0%)	8
	MF	7 (43.8%)	6 (37.5%)	0 (0%)	3 (18.8%)	16
	Other	6 (17.6%)	13 (38.2%)	4 (11.8%)	11 (32.4%)	34
HCT-CI/Age (*n*)	0	5 (33.3%)	6 (40.0%)	2 (13.3%)	2 (13.3%)	15
	1	18 (25.0%)	19 (26.4%)	10 (13.9%)	25 (34.7%)	72
	2	14 (33.3%)	12 (28.6%)	6 (14.3%)	10 (23.8%)	42
	3	4 (21.1%)	6 (31.6%)	3 (15.8%)	6 (31.6%)	19
	4	6 (35.3%)	4 (23.5%)	4 (23.5%)	3 (17.6%)	17
	5	0	5 (100%)	0	0	5
	6	0	1 (100%)	0	0	1
	Missing	1 (12.5%)	2 (25%)	1 (12.5%)	4 (50%)	8
DRST-EBMT Score (*n*)	0	0	0	0	0	0
	1	0	1 (100%)	0	0	1
	2	2 (18.2%)	3 (27.3%)	3 (27.3%)	3 (27.3%)	11
	3	6 (22.2%)	11 (40.7%)	2 (7.4%)	8 (29.6%)	27
	4	12 (35.3%)	7 (20.6%)	6 (17.6%)	9 (26.5%)	34
	5	12 (34.3%)	10 (28.6%)	6 (17.1%)	7 (20%)	35
	6	9 (25.7%)	11 (31.4%)	4 (11.4%)	11 (31.4%)	35
	7	3 (15.0%)	7 (35.0%)	3 (15.0%)	7 (35%)	20
	8	1 (33.3%)	1 (33.3%)	1 (33.3%)	0 (0%)	3
	Missing	3 (25%)	4 (33.3%)	1 (8.3%)	4 (33.3%)	12
Cases of aGvHD (*n*)		31 (26.7%)	41 (35.3%)	21 (18.1%)	23 (19.8%)	116
Cases of cGvHD (*n*)		17 (38.6%)	12 (27.3)	6 (13.6%)	9 (20.5%)	44
Days in hospital for transplant		32 (25–128)	35 (23–188)	37 (26–283)	37 (26–169)	35 (23–283)
Follow-up of survivors (months)		12 (7.5–12)	12 (7.2–12)	12 (9.1–12)	12 (7.3–12)	12 (7.2–12)
Body mass index (kg/m^2^)		26.6 (17.6–44.4)	27.5 (18.2–39.1)	23.8 (16.2–46.3)	22.9 (16.6–33.5) *	25.9 (16.2–46.3)
Muscle mass per height square (kg/m^2^)		10.8 (7.7–15.8)	11.1 (7.2–13.5)	9.6 (7.3–12.8)	9.6 (6.2–12) *	10.4 (6.2–15.8)
Grip strength (kg)		35.9 (22.4–64.1)	39 (22.1–71.2)	36.2 (16.9–49.8)	33.3 (16.2–49.4)	35.2 (16.2–71.2)
Z-Score grip strength ()		0.6 (−0.6–2.9)	0 (−0.7–2.9)	−0.7 (−1.7–2.2) **	−0.9 (−3.5–0.8) **	−0.2 (−3.5–2.9)
Peak power per body mass (W/kg)		1.7 (0.9–3.1)	1.3 (0.7–2.6) *	1.6 (0.8–3)	1.4 (0.7–2.2) *	1.5 (0.6–3.1)
VO2peak per body mass (mL/min/kg)		21.4 (13.1–31)	14.7 (8.3–27.6) **	18 (7.8–31.1) *	16.7 (9.3–28.9) **	17.6 (7.8–31.1)
VO2peak percentage of normal value (%)		93.8 (80–134.2)	64 (37.1–79) **	78 (26.8–101.8) **	58.1 (26.7–86.6) **	72 (26.6–134.2)

Data are counts (*n*) or represent median (min-max). * indicates *p* < 0.05 and ** indicates *p* < 0.01 (one-way ANOVA, Tukey post-hoc-test) vs. Group 1. Note: Z-Score grip strength was calculated using the normal values from Dodds et al., 2014 [21]. MDS: Myelodysplastic syndrome, AML: Acute myeloid leukemia, ALL: Acute lymphatic leukemia, CML: Chronic myeloid leukemia, MF: Myelofibrosis. HCT-CI/age: Hematopoietic cell transplantation-specific comorbidity index adjusted for age. DRST-EBMT: German register for stem cell transplantation-European bone marrow transplantation risk score. aGvHD: acute Graft versus Host Disease. cGvHD chronic Graft versus Host Disease. Note that GvHD data relates to the 170 patients actually transplanted.

## Data Availability

The data underlying this article will be shared on reasonable request to the corresponding author.

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
