# Peer review of "Sarcopenia Screening Allows Identifying High-Risk Patients for Allogenic Stem Cell Transplantation"

_cancers, 2021, doi:10.3390/cancers13081771_

Round 1

Reviewer 1 Report

Allogenic stem cell transplantation (aSCT) is the only potentially curative treatment for high-risk hematological diseases. Despite advancements in supportive measures, aSCT outcome is still affected by considerable transplant-related mortality. The authors implemented a new sarcopenia assessment prior to aSCT to evaluate its predictive capability for all-cause and non-relapse-mortality. All patients initially scheduled for aSCT within a 25-month period were screened during pre-transplantation-routine for muscle mass, grip strength, and aerobic capacity (AC) by measuring peak oxygen uptake (VO2peak). One-hundred-seventy-eight patients were included and rated as normal (n=48), impaired aerobic capacity (n=56), pre-sarcopenic (n=26), sarcopenic (n=27), and severe sarcopenic (n=22) without significant age-differences between groups. Patients presenting with sarcopenia showed a significant 3-fold increase in all-cause and non-relapse mortality compared to patients with normal screening results. AC showed to be the strongest single predictor with a more than 2-fold increase of mortality for low AC. The author conclude that risk stratification based on combination of muscle mass, grip strength, and AC allowed identifying the high-risk subgroup in patients undergoing aSCT. The integration of such screening routine into the pre-transplantation workup would enable clinicians to prescribe individual and appropriate training programs for each patients. I think that this study excellent and highly intriguing for the clinicians in this field. This study is for acceptable for publication in Cancers with one minor revision.

  1. I can’t discriminate the longitudinal and similar line (dot) of Group1-4 in Figure 2A and 3A. Please revise more clearly by different mark (circle, triangle, or square with black and white).

Reviewer 2 Report

Kirsten et al. present an interesting and well-done prospective analysis about muscle strength or aerobic capacity with the outcome of allogeneic stem cell transplantation. The paper is highly interesting for the transplant community.

However, they are some minor aspects in the paper that have to be addresses before considering the manuscript for publication:

  • In the title and in the paper, the author mention “high risk patients”. Even if the main part of the readers know that it means high risk for outcome parameters like survival or non-relapse mortality, etc. this has to be defined and clarified in the manuscript.
  • The duration of hospitalization for transplantation of the different patient group would be an interesting information and probably easy to obtain. In addition, an information about the frequency of transfer of the patient to the intensive care unit is an important information and will give more information about the higher non-relapse mortality. What were the main reasons for mortality?
  • Was there a relation between pretreatment and patient groups defined her? Patients with an induction treatment are probably at higher risk for loss of muscle strength.
  • The analysis for muscle strength and aerobic capacity was done in a median about 1 month before transplantation. Was any intensive treatment administered to some of the patients between the measurements and transplantation?
  • Was there any treatment with corticosteroids administered to the patients before transplantation? Probably not to patients with myeloid diseases, but perhaps in ALL patients. And was there an effect of steroid treatment after transplantation an additional risk factor for non-relapse mortality? The combination of steroids (mostly indicated for GVHD) and preexisting sarcopenia is perhaps (and probably) a bad combination. The author can analyze this or if numbers are too small, they must discuss this point.
  • I suppose that most data considered for statistical analysis had not a normal distribution. So, author must consider calculating with “median” instead of “mean”. Please discuss this with a statistician.
  • The author defined a HCT-CI > 2 only for 42 of 178 patients. This seems exceptionally low to me if all patients transplanted in this center have been included in the study. In Germany, around 40 % of the patients transplanted have an HCT-CI score > 2.

Minor points:

Line 79: “shownKelsey“ à “shown by Kelsey” I suppose.

Line 219: HR is 129, not logical.

Reviewer 3 Report

In the manuscript the author described how sarcopenia could be a risk factor in patients who received allogenic stem cell transplantation. 

Authors provided a quite numerous cohort of patients however data are difficult to interpretate. I would suggest authors to deeply edit the text in order to present in a more schematic and clear way their data and their conclusions. Moreover, the results should be presented in a way that it is clear the statistic outcome and the significance of the obtained numbers. The text a bit confused and not always easy to follow. Considering that this is a descriptive manuscript the writing should be more accurate.

Round 2

Reviewer 2 Report

The authors answered to all questions of the comments and the manuscript has been improved significantly.

Reviewer 3 Report

Authors addresses my points.